# Identification of a Specific Inhibitor of Human Scp1 Phosphatase Using the Phosphorylation Mimic Phage Display Method

**Takuya Yoshida, Kazuki Yamazaki, Shunta Imai, Akinori Banno, Atsushi Kaneko, Kazuhiro Furukawa and Yoshiro Chuman ***

Department of Chemistry, Faculty of Science, Niigata University, 8050 2-no-cho, Ikarashi, Nishi-ku, Niigata 950-2181, Japan

* Correspondence: chuman@chem.sc.niigata-u.ac.jp; Tel./Fax: +81-25-262-6160

**Abstract:** Protein phosphatases are divided into tyrosine (Tyr) phosphatases and serine/threonine (Ser/Thr) phosphatases. While substrate trapping mutants are frequently used to identify substrates of Tyr phosphatases, a rapid and simple method to identify Ser/Thr phosphatase substrates is yet to be developed. The TFIIF-associating component of RNA polymerase II C-terminal domain (CTD) phosphatase/small CTD phosphatase (FCP/SCP) phosphatase family is one of the three types of Ser/Thr protein phosphatases. Defects in these phosphatases are correlated with the occurrence of various diseases such as cancer and neuropathy. Recently, we developed phosphorylation mimic phage display (PMPD) method with $AlF_4^-$, a methodology to identify substrates for FCP/SCP type Ser/Thr phosphatase Scp1. Here, we report a PMPD method using $BeF_3^-$ to identify novel substrate peptides bound to Scp1. After screening peptide phages, we identified peptides that bound to Scp1 in a $BeF_3^-$-dependent manner. Synthetic phosphopeptide BeM12-1, the sequence of which was isolated at the highest frequency, directly bound to Scp1. The binding was inhibited by adding $BeF_3^-$, indicating that the peptide binds to the active center of catalytic site in Scp1. The phosphorylated BeM12-1 worked as a competitive inhibitor of Scp1. Thus, PMPD method may be applicable for the identification of novel substrates and inhibitors of the FCP/SCP phosphatase family.

**Keywords:** Scp1; FCP/SCP phosphatase family; Ser/Thr protein phosphatase; inhibitor; peptide; phage display; PMPD method

## 1. Introduction

Protein phosphorylation is the most widespread type of post-translational modification. Phosphorylation on Ser/Thr residues is regulated rigidly by Ser/Thr protein kinases and Ser/Thr protein phosphatases [1]. Scp1 belongs to the TFIIF-associating component of RNA polymerase II C-terminal domain (CTD) phosphatase/small CTD phosphatase (FCP/SCP)-type Ser/Thr protein phosphatases, which is one of the three types of the Ser/Thr protein phosphatase family. The other types of Ser/Thr protein phosphatases are phosphoprotein phosphatases (PPP) and metal-dependent protein phosphatases (PPM) [2]. FCP/SCP phosphatases including Scp1 share the same catalytic domain architecture, a DXDX(T/V) motif. This divalent metal ion coordination site corresponds to the signature sequence found in a family of structurally related metal-dependent phosphohydrolases and haloacid dehalogenases [3]. $Mg^{2+}$ is essential for FCP/SCP phosphatase activity, and the catalytic pocket of Scp1 forms a negatively charged surface owing to the cluster of conserved aspartic acid residues in a DXDX(T/V) motif used for $Mg^{2+}$ coordination and catalysis [4]. Scp1, also known as CTDSP1 or NLI-IF, was originally identified as a Ser/Thr protein phosphatase for C-terminal domain

(CTD) of the largest subunit in RNA polymerase II (RNA pol II) [5,6]. Recently, more proteins including c-Myc, Twist, Akt, PML and Smad have also been identified as substrates for Scp1 suggesting that Scp1 functions as an important regulator of the tumor suppression pathway and the BMB/TGFβ signaling pathway, in addition to regulating the transcription by RNA pol II [7–11]. Scp1 has shown a binding preference for the PX(S/T)P sequence of its substrates including the CTD. However, several of the other reported substrates of Scp1 do not contain this sequence in their dephosphorylation sites [12,13]. Thus, the research on the identification of substrates for Scp1 during the last decade suggests that Scp1 dephosphorylates a variety of substrates and provides various different functions. For further understanding of the biological functions of Scp1, novel methods to identify the substrate sequences or motifs of Scp1 must be developed.

Repressor element 1 (RE-1)-silencing transcription factor (REST) is known as the repressor protein transcription factor that induces the neuronal gene silencing [14]. Scp1 is widely expressed in cervical tissues, mesenchymal tissues, ectodermal tissues, and undifferentiated neuroepithelial cells at embryonic stages and high expression of Scp1 is observed in skeletal muscles, when compared to the brain. The parallel exclusion of Scp1 and REST/NRSF in differentiated neural tissues was expected to have a function in neuronal gene silencing, and a chromatin immunoprecipitation (ChIP) study revealed that the REST interacted with Scp1 [15]. Recently, it has been reported that REST is a bona fide substrate for Scp1 *in vivo* and that Scp1 stabilized REST protein level by protecting REST from degradation [16]. In addition, dysregulation of REST has been implicated in multiple diseases, including cancer, and the upregulation of REST was observed in several cases of mesenchymal and glioma tumors in addition to brain tumors including medulloblastomas and neuroblastomas [15,16]. Thus, REST protein, which is dephosphorylated and stabilized by Scp1, has oncogenic roles in several types of cells, suggesting that inhibitors specific to Scp1 may act as good anti-cancer drugs through the attenuation of REST functions in oncogenesis. Therefore, the identification of Scp1 inhibitors is desired, not only to develop useful drugs for cancer, but also to clarify the biological functions of Scp1. However, only a few studies have been reported on the specific inhibitors for SCP/FCP type Ser/Thr protein phosphatases including Scp1 [17–19]. Zhang et al. have reported that rabeprazole worked as a specific inhibitor for Scps including Scp1, Scp2 and Scp3 but did not inhibit its close family members Fcp1 and Dullard or bacteriophage λ Ser/Thr phosphatases [17]. The high-resolution crystal structure of rabeprazole-bound Scp1 revealed that the small compound bound to the hydrophobic binding pocket adjacent to the Scp1 active site. However, rabeprazole is also an antiulcer drug used to treat gastroesophageal reflux disease (GERD) through the inhibition of $H^+/K^+$ ATPase by forming a covalent bond with the active site cysteine of $H^+/K^+$ ATPase [20]. The drug's ability to inhibit different targets highlights its risk to cause serious side effects. This justifies the importance to screen and identify inhibitors targeting the active site of individual enzymes, in order to develop specific inhibitors.

In order to identify the substrates/inhibitors targeting the active site of an enzyme, it is essential to understand the catalytic mechanisms of the enzyme. The substrates of Tyr phosphatase, which is one of the two major protein phosphatase families, have been confirmed so far by using substrate-trapping mutants, in which the active-site cysteine residue was replaced to accumulate the covalent catalytic intermediate between enzyme and substrate and for subsequent identification of the associated substrate. Such substrate mutants have not been identified in the Ser/Thr phosphatase family because they hydrolyze phosphate esters in a single step and do not form a covalent catalytic intermediate [21]. Wu and coworkers reported a substrate identification method for the Ser/Thr phosphatase family, an easy-to-use tool to identify the substrates for PPP type Ser/Thr phosphatase PP1 using the fusion protein of hypoactive PP1 mutant with the regulatory interactors of protein phosphatase one (PIPPO) [22]. Recently, we reported the development of a novel methodology, termed phosphorylation mimic phage display (PMPD), to identify the substrates for FCP/SCP type Ser/Thr phosphatase Scp1 using peptide phage display libraries with $AlF_4^-$ [23]. These methods are expected to give new insights to understand the biological functions for Ser/Thr phosphatases through the identification of substrates/inhibitors targeting the individual enzymes of the Ser/Thr phosphatase family.

A number of cocrystallizations of enzyme with metal fluorides have clarified that $AlF_4^-$ and $BeF_3^-$ imitate either a phosphoryl or a phosphate group in a variety of enzymes, such as ATPases, GTPases, kinases, mutases, phosphohydrolases, and phosphatases [24,25]. FCP/SCP phosphatase crystal structures have also been solved with aluminum fluoride ($AlF_4^-$) or beryllium fluoride ($BeF_3^-$), in which $AlF_4^-$ complex resembles the proposed pentacoordinate phosphorane transition state of the hydrolysis reaction and $BeF_3^-$ complex adduct adopts the tetrahedral geometry resembling the phosphoaspartate intermediate [26,27]. We have focused on the role of $AlF_4^-$ as a phosphate mimic and developed the PMPD method using peptide phage display with $AlF_4^-$ as a useful method to identify the substrate candidate peptides for Scp1 phosphatase [23]. Both $AlF_4^-$ and $BeF_3^-$ work as mimics of phosphoryl or a phosphate group, however the geometry of these metal fluorides in complex with enzymes are different from each other; $AlF_4^-$ forms a pentacoordinate phosphorene complex with the enzymes and $BeF_3^-$ forms tetrahedral adduct with catalytic aspartate side chains of the enzymes [26,27]. Kamenski et al. also reported that the phosphatase activity of both recombinant Scp1 and Fcp1 was essentially abolished by the $BeF_3^-$ although the inhibitory effect of $AlF_4^-$ was less severe [26]. These facts suggested the possibility that $BeF_3^-$ may exhibit stronger interaction to Scp1 than $AlF_4^-$ and that the screening of binding peptides by PMPD method with $BeF_3^-$ against Scp1 may provide new substrate/inhibitor candidates for Scp1 and give us new insights to clarify the biological function of Scp1 by comparing the identified substrate candidates by PMPD methods with $AlF_4^-$.

Here we report that the PMPD method using $BeF_3^-$ instead of $AlF_4^-$ identified novel substrate peptides against recombinant human Scp1 (rScp1).

## 2. Results

### 2.1. Specific Inhibition of rScp1 by $BeF_3^-$

$BeF_3^-$ has been known to work as a phosphate group-mimic and form a stable tetrahedral adduct with catalytic aspartate side chains, mimicking a labile phosphoaspartate intermediate in phosphatases and mutases of the haloacid dehalogenase (HAD) superfamily [26,28,29]. Crystallization of N terminus-truncated Scp1 with $BeF_3^-$ also revealed that the beryllofluoride worked as a phosphate mimic and inhibited the phosphatase activity [27]. We also reported $AlF_4^-$, which worked as phosphate mimic. It induced the transition state of FCP/SCP type phosphatase Fcp1, inhibited recombinant human full-length Scp1 (rScp1), and was used in the PMPD method to identify substrate candidates for the FCP/SCP phosphatase family [23]. In order to investigate whether $BeF_3^-$ could work as a phosphate mimic for rScp1 and PMPD method to identify substrate candidates for rScp1 like $AlF_4^-$, we analyzed the effects of various inhibitors of the phosphoryl-transfer reaction on the activity of rScp1 (Figure 1A). Phosphatase activity of rScp1 using CTD(5pS), which is a phosphorylated peptide at Ser5 of CTD-derived heptad sequence, was essentially abolished by $BeF_3^-$. In contrast, neither $Be^{2+}$ nor $F^-$ alone had a strong effect on the activity of rScp1. These findings indicated that $BeF_3^-$ was located at the active center of rScp1, forming a Scp1–$Mg^{2+}$–$BeF_3^-$ complex that mimicked the acrylphosphate intermediate in Scp1 identical to that observed for FCP/SCP-type Ser/Thr phosphatase Fcp1 [27]. This inhibitory activity mode of $BeF_3^-$ against rScp1 corresponds to that of $AlF_4^-$ which we reported earlier, suggesting that $BeF_3^-$, like $AlF_4^-$, can be used in the screening of substrate candidates for Scp1 phosphatase in combination with our PMPD method [23].

Prior to screening for Scp1 substrate candidates using the PMPD method, we examined the effects of $BeF_3^-$ on Scp1 activity in a 96-well microtiter plate coated with rScp1 (Figure 1B). rScp1 phosphatase activity inhibited by the addition of $BeF_3^-$ was not restored after single washing with buffer without $BeF_3^-$ and more than 12 washes were needed for 80% recovery of the full activity. However, rScp1 phosphatase activity that was inhibited by $AlF_4^-$ was restored almost completely after only single washing without $AlF_4^-$ [23]. These data indicated that $BeF_3^-$ could form a stable tetrahedral adduct with catalytic aspartate side chains of $AlF_4^-$.

Similar observations of inhibitory activities between $AlF_4^-$ and $BeF_3^-$ were reported in yeast Fcp1 and N-terminal truncated Scp1 [26]. These studies suggested that $BeF_3^-$, which forms stable Scp1–$Mg^{2+}$–$BeF_3^-$ complex, may be more applicable to be used in the PMPD method to identify the substrate candidates for human Scp1 by optimal elution with elution buffer lacking $BeF_3^-$.

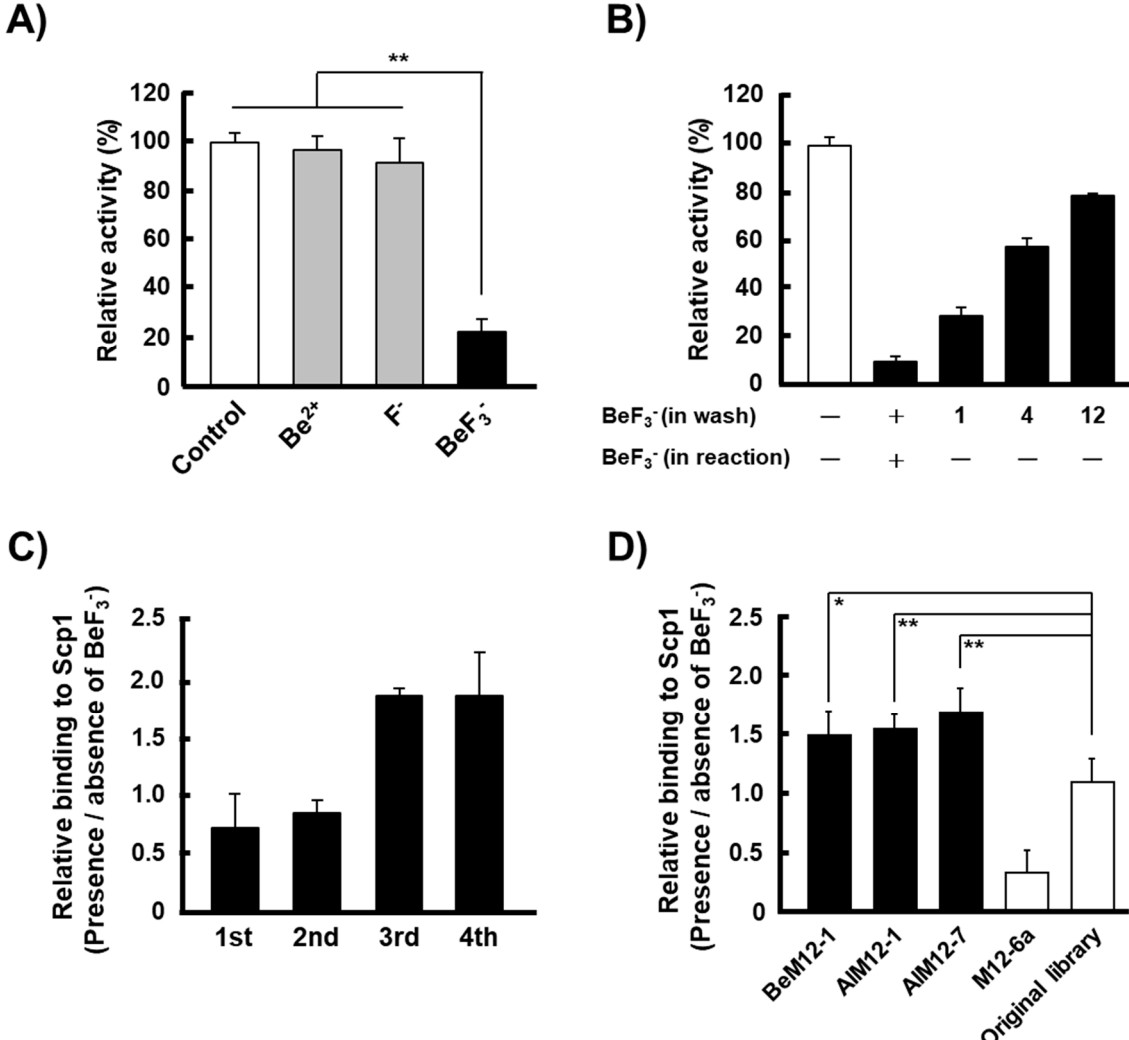

**Figure 1.** Phosphorylation mimic phage display (PMPD) method for recombinant human full-length Scp1 (rScp1) using $BeF_3^-$. (**A**) Specific inhibition of rScp1 activity by $BeF_3^-$. The data are means ± s.d. (** $p < 0.01$, *t*-test; N = 9). The inhibitors were added to reaction mixtures at the following concentrations: 1 mM NaF, 0.05 mM $BeSO_4$, and a mixture of 1 mM NaF and 0.05 mM $BeSO_4$ (to produce the $BeF_3^-$ *in situ*). (**B**) Rescue of rScp1 activity after the removal of $BeF_3^-$. The data are means ± s.d. (N = 3). The number on the horizontal axis are the number of washes using a maleate buffer without $BeF_3^-$. The activity of coated rScp1 was rescued after removing $BeF_3^-$ by repeated washing, although rScp1 was inhibited the phosphatase activity in the presence of $BeF_3^-$ in the phosphatase reaction buffer. (**C**) The PMPD method was performed using rScp1 fixed onto the ELISA plate to screen a randomized Ph.D.-M12 library. The relative binding affinity of phages recovered in each round for rScp1 was determined by the presence/absence of $BeF_3^-$. The data are means ± s.d. (N = 4). (**D**) $BeF_3^-$-specific binding of BeM12-1, AlM12-1 and AlM12-7, which are phages isolated from Ph.D.-M12 phage library, were analyzed for rScp1 in the presence/absence of $BeF_3^-$. M12-6a phage clone and original Ph.D.-M12 phage library were used as controls. The data are means ± s.d. (* $p < 0.1$, ** $p < 0.01$, *t*-test; N = 9).

### 2.2. Screening of Scp1 Substrate Candidates Using the PMPD Method with $BeF_3^-$

In order to screen for Scp1 substrate candidates using the PMPD method with $BeF_3^-$, we used a linear dodecapeptide Ph.D.M-12 library containing ~$10^{10}$ pfu individual recombinant clones. The library was screened using four rounds of panning with rScp1 immobilized on the surfaces of three wells in a 96-well microtiter plate containing $Mg^{2+}$ and $BeF_3^-$. After each round of panning, phages that were bound to rScp1 were eluted with maleate buffer lacking $BeF_3^-$. The binding activities of the phages after the fourth round of biopanning against rScp1 in the condition of the presence or absence of $BeF_3^-$ are shown in Figure 1C. Compared with the initial library, the phages after the third biopanning clearly exhibited increased binding to rScp1 in the presence of $BeF_3^-$, suggesting that PMPD method with $BeF_3^-$ was able to identify the substrate candidates of rScp1 like that with $AlF_4^-$. The phages after fourth panning were cloned and the sequences of the identified peptides are shown in Table 1. Among the 26 sequenced clones, three clones were isolated with multiple frequency although six other clones were detected once. BeM12-1 clone was isolated by PMPD method with $BeF_3^-$ targeting rScp1 at highest frequency. Two other clones, AlM12-1 and AlM12-7 previously identified by PMPD method with $AlF_4^-$ were also isolated in this study. These results suggested that both $BeF_3^-$ and $AlF_4^-$ may work as phosphate mimic and occupied the phosphorylated group binding sites in rScp1 although conformation of $BeF_3^-$ with Scp1 was different from that of $AlF_4^-$ in Scp1. In addition, all three clones with multiple frequency contained Ser or/and Thr residues with phosphorylation site in their sequence. These data suggested that Ser/Thr residues in the peptides isolated by the PMPD method will be powerful targets of Scp1 phosphatase.

**Table 1.** Sequences of the rScp1 binding clones.

| Name | Sequence | Frequency |
|---|---|---|
| BeM12-1 | TAKYLPMRPGPL | 12 |
| AlM12-1 | DYHDPSLPTLRK | 5 |
| AlM12-7 | SALPWFWSMDPS | 3 |
| Others | | 6 |
| | Total | 26 |

Peptides displayed on the phage clones were frequently selected by screening with rScp1 (BeM12-1, AlM12-1 and AlM12-7). BeM12-1 is obtained by the PMPD method with $BeF_3^-$. AlM12-1 and AlM12-7 are the sequence obtained by the PMPD method not only with $BeF_3^-$ but also $AlF_4^-$. Others are phage clones selected only once. Underlines indicate Ser and Thr residues, which are putative targeted sites by Scp1.

### 2.3. Ion-Specific Binding of Isolated Peptide-Displayed Phages to rScp1

Three clones isolated with multiple frequencies by the PMPD method using $BeF_3^-$ contained Ser or/and Thr residues in their sequences. To examine the binding properties of phages which displayed these peptides, detailed ELISA analyses were performed in the presence/absence of $BeF_3^-$. All of these phages showed increased relative binding to rScp1 in the presence of $BeF_3^-$ when compared to the absence of $BeF_3^-$ (Figure 1D). On the other hand, such an increase in binding affinity of rScp1 to $BeF_3^-$ were neither observed in the original M12 phage library nor M12-6a clone, which was reported as phage clone that bound to rScp1 in a metal fluoride-independent manner [23]. These data confirmed that the three peptides displayed on isolated phages, which were identified at multiple frequencies by the PMPD method using $BeF_3^-$ in this study, are powerful candidates to act as substrates of Scp1.

### 2.4. Phosphatase Activity of rScp1 against Phosphorylated Peptides Identified by the PMPD Method

To confirm whether the peptides isolated by the PMPD method with $BeF_3^-$ worked as substrates of Scp1, we synthesized the top two phosphorylated peptides that were frequently identified as Scp1-substrate candidates, BeM12-1 and AlM12-1. AlM12-1 contains two phosphorylated sites, Ser at position 6, and Thr at position 9. We synthesized dephosphorylated peptide, AlM12-1(6pS, 9pT) and monophosphorylated peptides, AlM12-1(6pS) and AlM12-1(9pT) (Table 2). In addition, we synthesized

two control peptides, the first being a universal peptide, which is commonly used as a standard phosphorylated substrate of Ser/Thr protein phosphatases, and the second being CTD(5pS) peptide, which is derived from endogenous Scp1-substrate, CTD phosphorylated at Ser5 [30–33].

To confirm whether the identified peptides work as the substrate for Scp1, we firstly carried out the phosphatase assay of the phosphorylated peptide analogues for AlM12-1, which were identified by the PMPD method using both $BeF_3^-$ and $AlF_4^-$. The phosphatase analysis revealed that rScp1 dephosphorylated AlM12-1(6pS, 9pT) and AlM12-1(6pS) but not AlM12-1(9pT), suggesting that phosphorylated Ser residue at the position 6 of AlM12-1 is the target site for Scp1 (Figure 2A). On the other hand, rScp1 also did not dephosphorylate the universal phosphorylated peptide, which contains phosphorylated Thr residue and is commonly used as a substrate for Ser/Thr protein phosphatases. This suggested that Scp1 prefers the phosphorylated Ser residue to Thr residue and that the universal peptide, which is commonly used as a substrate peptide for many Ser/Thr phosphatase studies, does not work as a substrate for rScp1 (Figure 2A). In addition, the phosphatase activity of rScp1 against diphosphorylated peptide AlM12-1(6pS, 9pT) was higher than that against monophosphorylated peptide AlM12-1(6pS). Zhang and coworkers had previously reported that SPT(pS) core sequence of CTD is key to the Scp1 interaction, whereas additional residues present on the N- and C-termini provide only little Scp1 binding using doubly phosphorylated 14mer peptide, YSPTSPSY(pS)PT(pS)PS [4]. This discrepancy of the effects of flanking sequence on dephosphorylation by Scp1 suggested that the additional interaction may occur between Scp1 phosphorylated Thr9 residue of AlM12-1(6pS, 9pT) unlike phosphorylated Ser2 on CTD(5pS).

Next, we carried out the test for phosphatase activity of rScp1 against BeM12-1(1pT), a substrate candidate identified at the highest frequency by the PMPD method using $BeF_3^-$. Interestingly, like the universal peptide, BeM12-1(1pT) peptide was not dephosphorylated at all by rScp1, while CTD(5pS), the physiological substrate peptide of Scp1 was dephosphorylated (Figure 2B). These results exhibited that the BeM12-1(1pT) could not work as a substrate for Scp1.

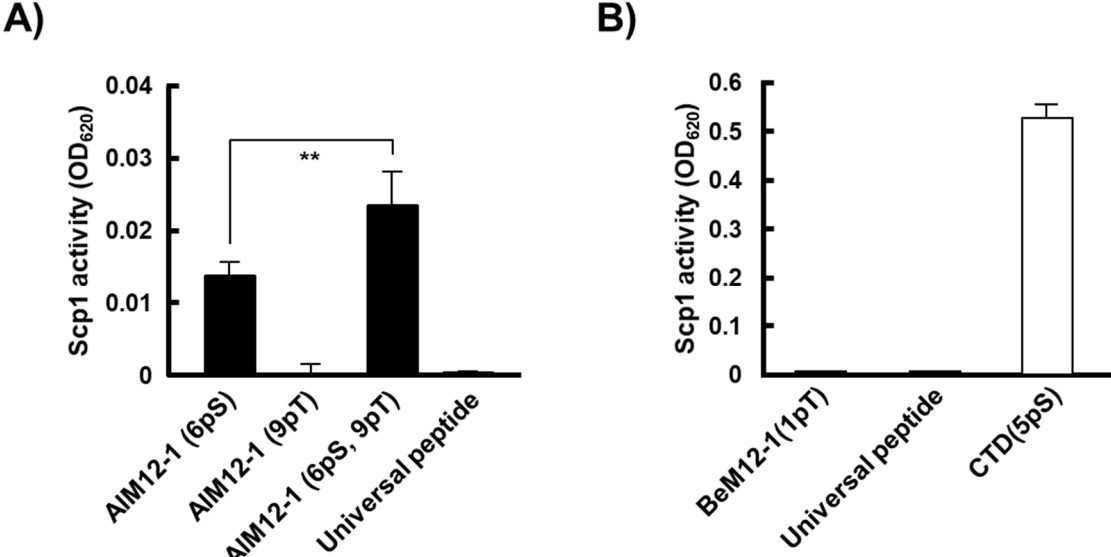

**Figure 2.** Phosphatase activity of rScp1 towards the synthesized phosphopeptides derived from the isolated phages by the PMPD method with $BeF_3^-$. (**A**) Phosphatase activity of rScp1 towards AlM12-1(6pS), (9pT) and (6pS, 9pT). The data are means ± s.d. (** $p < 0.01$, *t*-test; N = 9). The universal peptide was used as a control substrate for Ser/Thr protein phosphatases. (**B**) Phosphatase activity of rScp1 towards BeM12-1(1pT). The data are means ± s.d. (N = 12). C-terminal domain (CTD)(5pS) was used as a control substrate for rScp1.

**Table 2.** Synthetic phosphorylated peptides for phosphatase analyses by rScp1.

| Name | Sequence |
|------|----------|
| BeM12-1(1pT) | Ac-(pT)AKYLPMRPGPL-NH$_2$ |
| AlM12-1(6pS) | Ac-DYHDP(pS)LPTLRK-NH$_2$ |
| AlM12-1(9pT) | Ac-DYHDPSLP(pT)LRK-NH$_2$ |
| AlM12-1(6pS, 9pT) | Ac-DYHDP(pS)LP(pT)LRK-NH$_2$ |
| Universal peptide | H-RRA(pT)VA-OH |
| CTD(5pS) | Ac-SPSYSPT(pS)PS-NH$_2$ |
| p53(15P) | Ac-VEPPL(pS)QETFSDLW-NH$_2$ |
| Bio-BeM12-1(1pT) | bio-ε-(pT)AKYLPMRPGPL-NH$_2$ |

AlM12-1(6pS), (9pT), (6pS, 9pT) and BeM12-1(1pT) peptides are synthetic peptides recovered by the PMPD method with BeF$_3^-$. CTD(5pS) and universal peptide are used as control peptides. p53(15P) peptide was used as a substrate for PPM1D. Bio, biotin; ε, ε-aminocaproic acid; Ac, acetylation; pS, phosphorylated serine; pT, phosphorylated threonine. Underlines indicate putative sites targeted by rScp1.

## 2.5. Inhibition of BeM12-1(1pT) Binding to rScp1 by BeF$_3^-$

In order to examine whether phosphorylated BeM12-1(1pT) can bind to rScp1 directly, we also synthesized biotinylated BeM12-1(1pT), named as bio-BeM12-1(1pT), and subjected it to ELISA using avidin-HRP. As expected, bio-BeM12-1(1pT) bound to rScp1 in a dose-dependent manner, while it did not bind to BSA (Figure 3). Interestingly, the binding of bio-BeM12-1(1pT) to rScp1 was inhibited with the addition of BeF$_3^-$, suggesting that the bio-BeM12-1(1pT) interacts with the substrate binding site of rScp1 and the phosphorylation site of BeM12-1(1pT) may be located at the same position as of the BeF$_3^-$ binding site in Scp1. These data suggested that the BeM12-1(1pT) peptide may be a Scp1-specific inhibitor.

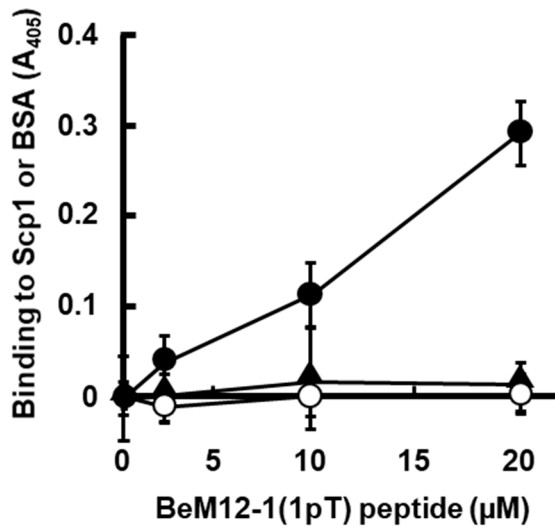

**Figure 3.** Binding BeM12-1(1pT) peptide to rScp1 in ELISA assay. The analysis of binding affinity of BeM12-1(1pT) peptide for rScp1 with/without BeF$_3^-$ on ELISA plate (filled circle, rScp1 without BeF$_3^-$; open circle, BSA without BeF$_3^-$; filled triangle, rScp1 with BeF$_3^-$). The data are means ± s.d. (N = 9).

## 2.6. Inhibitory Activity of BeM12-1(1pT) against rScp1

ELISA analysis using bio-BeM12-1(1pT) and BeF$_3^-$ suggested that the bio-BeM12-1(1pT) bound to the active center of rScp1 and inhibited the phosphatase activity. Inhibitory activity of BeM12-1(1pT) against rScp1 was evaluated using CTD(5pS) as a substrate. The dephosphorylation assay showed that the BeM12-1(1pT) peptide inhibited rScp1 phosphatase activity in a dose-dependent manner and the IC$_{50}$ value was 100.4 μM (Figure 4A).

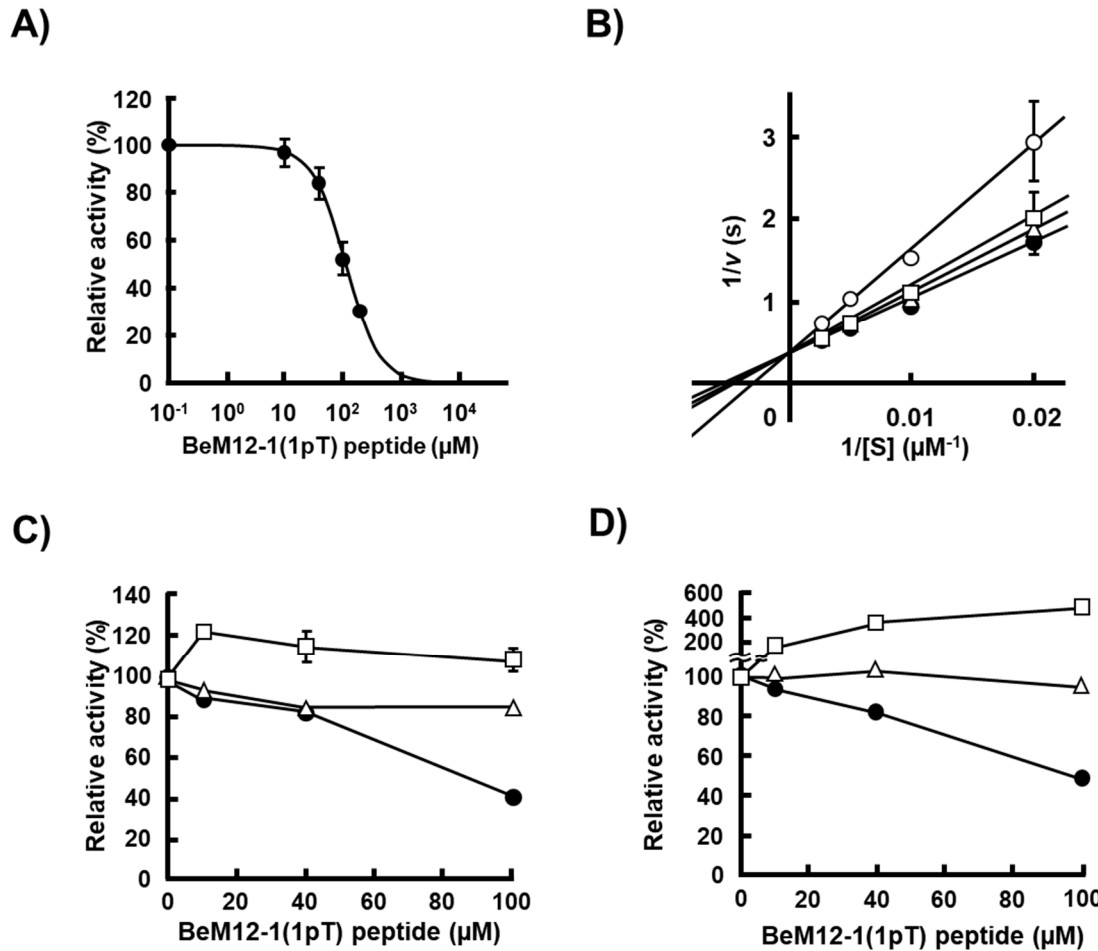

**Figure 4.** Inhibition of rScp1 phosphatase activity by BeM12-1(1pT). (**A**) Inhibitory activity of BeM12-1(1pT) peptide for rScp1. Inhibition assay of rScp1 activity using CTD(5pS) as a substrate was detected using BIOMOL GREEN Reagent. Inhibition assay was performed in the presence of 10 mM $Mg^{2+}$ for 10 min at 37 °C. The data are means ± s.d. (N = 9). The solid line represents the best fit obtained by fitting using GraphPad Prism8. (**B**) Determination of inhibitory mode by BeM12-1(1pT) peptide for rScp1 using CTD(5pS) as a substrate. Concentrations of BeM12-1(1pT) peptide: 0 μM (filled circle), 20 μM (open triangle), 100 μM (open square) and 200 μM (open circle). The vertical axis is the inverse of the velocity and the horizontal axis is the inverse of the substrate concentration in the enzyme reaction. Global fitting analysis suggested competitive inhibition mode (GraphPad Prism8). (**C**) Specific inhibition of rScp1 activity by BeM12-1(1pT) peptide using pNPP as a substrate; rScp1 (filled circle), PPM1D (open triangle) and PPM1A (open square). (**D**) Specific inhibition of rScp1 activity by BeM12-1(1pT) peptide using CTD(5pS) for rScp1 (filled circle), universal peptide for PPM1A (open square) and p53(15P) for PPM1D (open triangle).

Next, we investigated the inhibitory mechanism of BeM12-1(1pT) to rScp1. Figure 4B shows the double-reciprocal plot of rScp1 dephosphorylation of CTD(5pS) in the absence or presence of BeM12-1(1pT) peptide. The inverse of *v*, the rate of the Scp1-mediated reaction, is shown on the vertical axis, while the inverse of the substrate concentration is shown on the horizontal axis. Fitting of the data using the Lineweaver–Burk equation provided the maximum reaction velocity $V_{max}$ from the inverse of the intercept of the vertical axis and the Michaelis constant $K_m$, which indicates the binding affinity of the substrate to rScp1 from the inverse of the intercept on the abscissa. The data show that the $V_{max}$ value was independent of the concentration of BeM12-1(1pT) peptide. These data indicated that the presence of the BeM12-1(1pT) peptide decreased the substrate binding affinity in rScp1 but not the rate of the reaction (Figure 4B). Thus, these results demonstrated that the BeM12-1(1pT) acted as

a competitive inhibitor of rScp1 and the $K_i$ value was 242.6 μM. This result corresponded to that of the binding assay of bio-BeM12-1(1pT) with $BeF_3^-$ in which the BeM12-1(1pT) bound to the active center of rScp1 (Figure 3).

Finally, we examined the specificity of BeM12-1(1pT) to other types of Ser/Thr protein phosphatases, PPM1A and PPM1D. These phosphatases belong to the PPM type of protein phosphatase and work as a monomer like FCP/SCP-type phosphatase including Scp1. BeM12-1(1pT) peptide inhibited rScp1 but not PPM1A or PPM1D (Figure 4C). A similar phenomenon of enzymatic specificity of BeM12-1(1pT) was observed in the inhibitory analysis using CTD(5pS) phosphorylated peptide as a substrate, while BeM12-1(1pT) was dephosphorylated by PPM1A (Figure 4D). These data suggested that BeM12-1(1pT) worked as a specific inhibitor of Scp1.

## 3. Discussion

Ser/Thr protein phosphatases are divided into three subtypes: PPP, PPM and FCP/SCP phosphatase families. A rapid and simple method to identify substrates of Ser/Thr phosphatases has hardly been reported. Fusion protein of PPP-type Ser/Thr protein phosphatase PP1 with its regulatory interactors has been reported to serve as substrate trapping mutants to identify substrates for PPP [22]. However, this method is not feasible for identifying the substrates of the other two subtypes of Ser/Thr protein phosphatases, PPM and FCP/SCP phosphatases, because PPP type phosphatase PP1 works as a holoenzyme whereas PPM and FCP/SCP phosphatase family work as a monomers. Recently we have reported a novel methodology, termed phosphorylation mimic phage display (PMPD), to identify the substrates for FCP/SCP-type Ser/Thr phosphatase Scp1 using peptide phage display libraries with $AlF_4^-$ [23]. In this study, three peptide sequences were identified as Scp1-binding candidates in a $BeF_3^-$-dependent manner. BeM12-1 was identified when screened by only PMPD method with $BeF_3^-$, whereas both AlM12-1 and AlM12-7 peptides were identified previously by the PMPD method with $AlF_4^-$. Phosphatase analyses using the synthetic phosphopeptides of AlM12-1 and BeM12-1 revealed that AlM12-1(6pS) was dephosphorylated by rScp1, however BeM12-1(1pT) inhibited Scp1 activity. BeM12-1(1pT) peptide directly bound to the rScp1 and the binding was inhibited by the addition of $BeF_3^-$, indicating that the peptide binds to the active center of catalytic site in Scp1. Interestingly Lineweaver–Burk plot analysis suggested that the BeM12-1(1pT) peptide works as a competitive inhibitor of Scp1. Collectively, these data suggest that PMPD method may be applicable for the identification of both novel substrates and inhibitors of the FCP/SCP phosphatase family.

Three phage clones with peptides that acted as Scp1 substrates were isolated by using the PMPD method with $BeF_3^-$; BeM12-1 clone was isolated at the highest frequency in comparison to the other two clones, AlM12-1 and AlM12-7, which were also isolated previously by the PMPD method with $AlF_4^-$. Phosphatase analysis revealed that rScp1 dephosphorylated AlM12-1(6pS) but not AlM12-1(9pT), suggesting that Scp1 prefers phosphorylated Ser to phosphorylated Thr as target residue. Most of the residues reported so far to be targeted by Scp1 are Ser residues including CTD(5pS) of RNA pol II; the hydrophobicity of Pro residue located next to phosphorylated Ser5 in CTD plays an important role in substrate recognition by Scp1 through the interaction of the substrates with the hydrophobic pocket next to the active center of Scp1 [4,8–10,27,34–36]. These facts indicated that the hydrophobicity or space occupation of the methyl group in Thr reside may impair the recognition of substrates by Scp1. In addition, regarding of the recognition of AlM12-1(6pS) by Scp1, the peptide did not contain the pSP motif which was observed in most Scp1-substrates reported previously, although it contained two Pro residues in the sequence. In addition, phosphatase activity analysis of double phosphorylated AlM12-1(6pS, 9pT) peptide suggested that the phosphorylation of Thr residue at position 9 enhanced the phosphatase activity of Scp1 against pSer at position 6, whereas it is known that the phosphorylated residues at the flanking region of phosphorylation site were not affected in CTD(5pS) [4]. The discrepancy of effects of flanking residues on phosphatase activity between AlM12-1(6pS) and CTD(5pS) suggested their differences in the binding mode to the active center pocket and flanking hydrophobic pocket in Scp1. The crystal structure of Scp1 mutated in active center,

such as Scp1 D96N mutant with AlM12-1(6pS) will give us new insights on the substrate recognition of Scp1 through the clarification of binding mode of the phosphorylated peptide and Scp1.

Homology search of the sequence of AlM12-1 peptide from protein database exhibited high similarity of the peptide to spectrin and tripartite motif-containing protein 46 (TRIM46); spectrin is also known as WASH complex subunit 5, which is related to Ritscher-Schinzel syndrome 1 (RTSC1) characterized by craniofacial abnormalities and cerebellar brain malformations, and TRIM46 controls neuronal polarity and axon specification, whereas the phosphorylation in the similar sequence has not been reported [37,38]. Both proteins play important roles in neurological signal transduction and function and disorders of these proteins are expected to cause neurological diseases, suggesting that the functions of these proteins may be regulated by Scp1, which also regulates the neurological system. Thus, future studies of regulation of these proteins by Scp1 may reveal new biological functions of Scp1 through the regulation of spectrin and TRIM46.

BeM12-1(1pT), derived from the peptide isolated by the PMPD method at the highest frequency, worked as a competitive inhibitor against rScp1, but not against other types of Ser/Thr phosphatases, PPM1A and PPM1D. Inhibitory activity of BeM12-1(1pT) correlated with preferential binding to the pSer in comparison to pThr as substrate for Scp1, as described above. BeM12-1(1pT) is the first peptide which interacted with the active center pocket of wild-type Scp1 directly and inhibited the phosphatase activity of Scp1. The phosphorylated peptide was dephosphorylated by PPM type phosphatase PPM1A and this coincides with the fact that pThr is the most often observed site for dephosphorylation by the most PPM1A-substrates reported so far [39–41]. In the future, the modification of peptide bonds such as pseudobonds will confer resistance against dephosphorylation by PPM1A for the BeM12-1(1pT) peptide and it may be applicable as a lead compound as a Scp1-specific inhibitor in *ex vivo* and *in vivo* analysis.

CTD phosphorylated at the Ser5 is the sole peptide which was reported to have a crystal structure with Scp1; however, in the X-ray crystal structure Scp1 mutant (D96N) was used. BeM12-1(1pT) peptide identified in this study can form a stable complex with wild-type Scp1 in the active center pocket, suggesting that the crystal structure of the phosphorylated peptide with wild-type Scp1 will give new insights on the substrate recognition mechanism of Scp1 and it will be useful for the development of Scp1 inhibitors to study the biological function of Scp1 *in vivo*.

Ser/Thr protein phosphatase is known to dephosphorylate not only pSer but also pThr, however, several Ser/Thr protein phosphatases are also known to show dephosphorylation preference to either pSer or pThr [42,43]. Notably, it is also known that the distribution of pSer and pThr in human cells is in a 5:1 ratio of pSer:pThr [44,45]. Phosphatase analyses using phosphorylated peptide analogues in this study showed that Scp1 may have a strong preference of pSer over pThr as its substrate. Studies focused on understanding pSer/pThr preference by Ser/Thr protein phosphatases will give us an understanding of the mechanism of pSer/pThr distribution in human cells and will be useful for the development of specific inhibitors against Ser/Thr protein phosphatases.

Recently, abnormal expression of REST, known as Scp1-targeted protein, has been reported to be observed not only in tumors but also in neurological diseases including Parkinson's and Huntington's disease [46–48]. This indicates the importance to develop a method for substrate identification for identification of specific Scp1 inhibitors. Rabeprazole is the first selective lead compound targeted against human Scp1 and the crystal structure of rabeprazole-bound Scp1 showed that the compound bound to the hydrophobic binding pocket close to the active center of Scp1 [17,49]. However, rabeprazole is also known to inhibit $H^+/K^+$ ATPase by forming a covalent bond with the active site cysteine and is used to treat gastroesophageal reflux disease (GERD). This highlights the importance of using rabeprazole with caution in clinical applications even though it is a good candidate as Scp1-specific inhibitor. Park and coworkers have reported the discovery of both competitive and allosteric Scp1 inhibitors through their two-track virtual screening procedure [18]. In their screening, they identified three allosteric and five competitive Scp1 inhibitors using a chemical database of 600,000 products. BeM12-1(1pT) peptide identified in this study is the first peptidyl inhibitor for Scp1

and other FCP/SCP type Ser/Thr protein phosphatases, in contrast to other Scp1-inhibitors derived from chemicals. However, its inhibitory activity is still not strong. Understanding of the nature of binding between wild-type Scp1 and BeM12-1(1pT) and further optimization of the peptide may lead to development of clinical inhibitors for Scp1 as anti-cancer drugs and therapeutic agents for neurological diseases including Parkinson's and Huntington's disease.

## 4. Materials and Methods

### 4.1. Materials

The bacteriophage Ph.D.-M12 library was purchased from New England BioLabs (NEB, Beverly, MA, USA). Anti-M13 bacteriophage antibody, anti-rabbit antibody conjugated horseradish peroxidase (HRP), avidin conjugated HRP were from Sigma-Aldrich (B7786-2ML, Saint Louis, MO, USA), Santa Cruz Biotechnology (sc-2313, Santa Cruz, CA, USA) and eBioscience (18-4100-94, San Diego, CA, USA), respectively. Human Scp1 cDNA was obtained from Open Biosystems (Huntsville, Al, USA). Phosphorylated peptides (CTD(5pS): Ac-SPSYSPT(pS)PS-NH$_2$ and universal peptide: H-RRA(pT)VA-OH, in which pS and pT indicate phosphorylated Ser and Thr residues, respectively) were chemically synthesized by Toray Research Center (Tokyo, Japan).

### 4.2. Synthesis of Scp1 Substrate Candidate Peptides

The synthesis of phosphorylated peptides (AlM12-1(6pS), (9pT), (6pS, 9pT), BeM12-1(1pT) and p53(15P)) with N-terminal acetylation or biotin labeling was performed by standard Fmoc (9-fluorenylmethyloxycarbonyl) solid phase peptide synthesis, using CLEAR amide resin (Nacalai Tesque, Kyoto, Japan). Following cleavage from the resin, peptides were precipitated with diethyl ether, vacuum dried, and purified by HPLC (using a GL Science Inertsil C8 column) with H$_2$O/0.1% (v/v) trifluoroacetic acid (TFA) (A) and acetonitrile/0.1% (v/v) TFA (B) as solvents. The peptides were eluted using the HPLC column and a linear gradient (5%–40% B for 35 min; flow rate 1 mL/min). The purified peptides were lyophilized and subjected to MS analysis using ESI mass spectrometry (Thermo Fisher Scientific, Bremen, Germany) to confirm the correct molecular weight. Concentrations were estimated by absorption using the extinction coefficient at 280 nm, which was based on Tyr content. Standard Fmoc AA and Fmoc phosphorylated AA were purchased from Novabiochem and D-biotin was from Nacalai Tesque.

### 4.3. In Vitro Phosphatase Assay of rScp1

rScp1 expression and purification were performed as described [23]. In order to analyze the rScp1 phosphatase activity, pNPP and phosphorylated CTD peptides were used as substrates. pNPP assay was carried out in maleate buffer (20 mM maleate pH 5.5, 30 mM MgCl$_2$) with 10 nM of rScp1 at 37 °C in 100 μL volume for 7 min. The reactions were stopped with 50 μL the solution (0.1 M Tris(hydroxymethyl)aminomethane, 2% SDS). Released *p*-nitrophenol was quantified by measuring absorbance at 410 nm. Phosphatase activity of rScp1 against phosphorylated peptides were measured in maleate buffer (20 mM maleate pH 5.5, 10 mM MgCl$_2$) with 10 nM of rScp1 at 37 °C in 50 μL volume. After 10 min reaction, the reactions were quenched by adding 100 μL of malachite green reagent (Enzo Life Sciences, Plymouth, PA, USA). The release of free inorganic phosphate was determined by measuring absorbance at 620 nm. To determine the rScp1 kinetic parameters $k_{cat}$ and $K_m$, the initial velocities ($v$) were measured at pNPP and phosphorylated peptide concentrations (S), and data were fitted to the Michaelis–Menten Equation (1) with fitting software KaleidaGraph 4.0 (Synergy Software, Reading, PA, USA).

$$v = (k_{cat}·[S])/(K_m + [S]) \tag{1}$$

In order to analyze the inhibitory of BeM12-1(1pT) against rScp1, PPM1A and PPM1D, CTD(5pS) peptide, Graph peptide and p53(15P) peptide were used as their peptidyl substrates, respectively,

in addition of pNPP as common substrate for protein phosphatases. In the inhibition assay using pNPP, BeM12-1(1pT) peptide at the concentrations of 0, 10, 40, or 100 μM was incubated with substrate/enzyme reaction solution. The assays with PPM1A or PPM1D were carried out in Tris buffer solution (50 mM Tris-HCl pH 7.5, 0.02% 2-mercaptoethanol, 0.1 mM glycol ether diamine tetraacetic acid, 10 mM MnCl$_2$) with 10 nM of PPM1A or PPM1D at 30 °C in 100 μL volume although the assay with rScp1 was carried out as described above. In the inhibition assay using the phosphorylated peptides as substrate, the reaction was performed in the same condition with the inhibitory assay using pNPP except for the volume of 50 μL reaction solution with 30 mM MgCl$_2$ instead of 100 μL with 10 mM MnCl$_2$. IC$_{50}$ value was determined with fitting software KaleidaGraph 4.0 (Synergy Software, Reading, PA, USA).

To determine the inhibition constant ($K_i$), data were fitted to the Michaelis–Menten Equation (2), where the initial velocities ($v$) was obtained by fitting the data with Equation (1) and $V_{max}$, $K_m$, [S] and [I] are the maximum velocity of enzyme reaction, the $K_m$ of the rScp1 for substrate, phosphorylation peptide concentrations and BeM12-1(1pT) peptide concentrations, respectively, using fitting software GraphPad Prism8 (GraphPad Software, Inc., San Diego, CA, USA).

$$v = (V_{max} \cdot [S])/\{K_m(1 + [I]/K_i) + [S]\} \tag{2}$$

### 4.4. Inhibition Assay of rScp1 by Inhibitors including BeF$_3$$^-$

The activity of Scp1 toward CTD(5pS) peptide in the absence or presence of BeF$_3$$^-$ was measured in maleate buffer with 10 nM Scp1 for at 37 °C for 10 min. The inhibitors were added to reaction mixtures in the following concentrations: 1 mM sodium fluoride (NaF), 0.05 mM beryllium sulfate (BeSO$_4$), and a mixture of 1 mM NaF and 0.05 mM BeSO$_4$ to produce BeF$_3$$^-$ *in situ*.

### 4.5. BeF$_3$$^-$ Inhibition and Reactivity Assay on ELISA Plate

The activity of Scp1 toward pNPP in the absence or presence of BeF$_3$$^-$ was measured in maleate buffer with a coating of Scp1 (2.0 μg) at 4 °C overnight in 100 μL volume on enzyme-linked immunosorbent assay (ELISA) plate. Then, the plate was blocked with 0.5% (w/v) bovine serum albumin (BSA) in maleate buffer at room temperature for 2 h in 200 μL volume. The wells were washed 1, 4, or 12 times with maleate BeF$_3$$^-$ buffer (20 mM maleate pH 5.5, 150 mM NaCl, 10 mM MgCl$_2$, 1.0 mM NaF, 0.05 mM BeSO$_4$). Then, the activity of rScp1 toward pNPP in the absence or presence of BeF$_3$$^-$ was measured on maleate buffer coated with rScp1 at room temperature for 7 min in 100 μL volume. To estimate the amount of pNPP dephosphorylated by rScp1, the absorbance at 405 nm was measured using microplate reader ChroMate 4300 (Awareness Technology Chromate, Palm City, FL, USA).

### 4.6. Screening of Phage Displayed Combinatorial Peptide Library against rScp1 with Mg$^{2+}$ and BeF$_3$$^-$

The Ph.D.-M12 Phage Display Libraries (New England BioLabs, Beverly, MA, USA) were screened using purified rScp1 (2.0 μg in 100 μL of 20 mM maleate pH5.5, 150 mM NaCl) immobilized on ELISA plate. After swirling to wet the whole surface of the well, the plate was incubated at 4 °C overnight. Then, the plate was blocked with 0.5% (w/v) BSA in maleate buffer at room temperature for 2 h, before adding approximately 1 × 10$^{10}$ plaque-forming units (pfu) of the phages in 100 μL of maleate blocking BeF$_3$$^-$ buffer. After incubating the plate at room temperature for 1 h, the wells were washed 30 times with maleate BeF$_3$$^-$ buffer to remove unbound phages. Then, phages that bind specific ions were eluted 4 times with 100 μL of maleate buffer without BeF$_3$$^-$. Eluants from the first two cycles were amplified and titered. After the third and fourth rounds of screening, the eluted phages were titered without amplification. Binding phage clones were randomly selected from the titration plate for further analysis.

*4.7. Binding Analysis of Isolated Phage Clones to rScp1 with Ions in ELISA*

Isolated phage clones ($10^{10}$ pfu) were added to rScp1 (2.0 μg) immobilized on the surface of a well from ELISA plate in maleate buffer with/without $BeF_3^-$ for 1 h at room temperature. After washing 30 times with maleate buffer with/without $BeF_3^-$, the bound phages were detected with anti-M13 antibody (Sigma) and anti-rabbit antibody conjugated HRP (SantaCruz). To estimate the content of bound phages, a solution of ABTS (2,2′-azinobis(3-ethylbenzothiazoline-6-sulfonic acid)-diammonium salt)/$H_2O_2$ was added to each well and the absorbance at 405 nm was measured using microplate reader ChroMate 4300 (Awareness Technology Chromate, Palm City, FL, USA).

*4.8. Binding Analysis of Biotinylated BeM12-1(1pT) Peptide to rScp1*

Biotinylated phosphorylation peptide (0, 2, 10, or 20 μM) was added to rScp1 (2.0 μg) immobilized on the surface of a well from ELISA plate in maleate buffer with/without $BeF_3^-$ for 2 h at room temperature. After washing six times using maleate buffer with/without $BeF_3^-$, the bound biotinylated peptide was detected with avidin conjugated HRP (Bioscience). To estimate the content of bound peptide, a solution of ABTS/$H_2O_2$ was added to each well and the absorbance at 405 nm was measured using microplate reader ChroMate 4300. In order to analyze the effect of $BeF_3^-$ in the binding of BeM12-1(1pT) to rScp1, the biotinylated peptide (20 μM) was added to rScp1 (2.0 μg) immobilized on the surface of a well from ELISA plate in maleate buffer with $BeF_3^-$ ($BeSO_4$ was added at 0, 0.01, 0.05, or 0.2 mM, and NaF was added at 0, 0.2, 1, or 4 mM, respectively) for 2 h at room temperature. After washing each concentration of $BeF_3^-$ six times using maleate buffer, the bound biotinylated peptide was detected with avidin conjugated HRP.

**5. Summary**

In this study, we report a PMPD method using $BeF_3^-$ instead of $AlF_4^-$ to identify the novel substrate peptides of Scp1. After screening peptide phages bound to rScp1, we identified that several clones bound to rScp1 in a $BeF_3^-$-dependent manner. Synthetic phosphopeptide BeM12-1(1pT) showed direct binding to the active center of rScp1 at the highest frequency. Kinetics analysis revealed that phosphorylated BeM12-1(1pT) did not work as a substrate but as a competitive inhibitor of rScp1. These data suggested that the PMPD method with $BeF_3^-$ may be applicable for the identification of both of novel substrates and inhibitors of FCP/SCP phosphatases family. In the future, studying the crystal structure of wild-type Scp1 with BeM12-1(1pT) will give new insights on the substrate recognition mechanism and on development of inhibitors for Scp1 which can be used to further understand the biological function of Scp1 *in vivo*.

**Author Contributions:** Conceptualization, T.Y., K.Y., and Y.C.; methodology, T.Y. and Y.C.; validation, T.Y., A.K. and Y.C.; formal analysis, T.Y.; investigation, T.Y., K.Y.,S.I., and A.B.; resources, A.K. and K.F.; data curation, T.Y., and K.Y.; writing—original draft preparation, T.Y.; writing—review and editing, T.Y., K.Y., S.I., A.K. and Y.C.; visualization, T.Y. and K.Y.; supervision, K.F. and Y.C.; project administration, Y.C.; funding acquisition, A.K. and Y.C.

**Funding:** This work was supported in part by a Grant-in-Aid for Scientific Research (C) (No. 15K05560) and (B) (No. 19H03512) (to Y.C.) from Japanese Society for the Promotion of Sciences, and Research Fellowships for Young Scientists from JSPS (No. 18J20422 to A.K.).

**Conflicts of Interest:** The authors declare no conflict of interest.

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
