# Peer review of "Identification of a Specific Inhibitor of Human Scp1 Phosphatase Using the Phosphorylation Mimic Phage Display Method"

_catalysts, doi:10.3390/catal9100842_

Round 1

Reviewer 1 Report

In this manuscript, the authors reported a rapid and simple method to identify substrates of Ser/Thr phosphatases. They used PMPD method with BeF3- instead of AlF4− to identify novel substrate peptides that can bind to the catalytic pocket of Scp1. Three peptide sequences were identified as Scp1-binding candidates. This method may be useful to identify novel substrates and inhibitors of the FCP/SCP phosphatase family. Two comments were suggested to the authors.

1) Line 102 to line 106

It may be better to use a picture to explain how AlF4- or BeF3- connect with the residues AA in the enzyme's activity pocket.

2) The authors should explain why BeF3- is better to identify novel substrate in PMPD method. Line 111 to line 122, these are results of the research, they should be moved to the discussion.

Reviewer 2 Report

Please find my comments about the results and the presentation below:

Figure 4 b: Please explain why there is a very large standard deviation in chart b? Materials and methods: Please describe in detail the conditions under which the HPLC analyzes were performed. Materials and methods:” After 7 min reaction, the reactions were quenched by adding 50 μL of stop solution (0.1 M Tris, 2% SDS).” Please change this sentence to remove the laboratory slang. What does it mean: The BeM12-1(1pT) peptide was added to reaction mixtures as the inhibitor at concentrations of 0, 10, 40 and 100 μ”? Line 479: The name of compounds should be written in small letter.
